# Detecting Photovoltaic Installations in Diverse Landscapes Using Open Multi-Source Remote Sensing Data

**Jinyue Wang** [1,2], **Jing Liu** [1,2,*] and **Longhui Li** [1,2]

1  Jiangsu Center for Collaborative Innovation in Geographical Information Resource Development and Application, Nanjing 210023, China
2  Key Laboratory of Virtual Geographic Environment, Nanjing Normal University, Ministry of Education, Nanjing 210023, China
*  Correspondence: liujingxitu@gmail.com

**Abstract:** Solar photovoltaic (PV) power generation is a vital renewable energy to achieve carbon neutrality. Previous studies which explored mapping PV using open satellite data mainly focus in remote areas. However, the complexity of land cover types can bring much difficulty in PV identification. This study investigated detecting PV in diverse landscapes using freely accessible remote sensing data, aiming to evaluate the transferability of PV detection between rural and urbanized coastal area. We developed a random forest-based PV classifier on Google Earth Engine in two provinces of China. Various features including Sentinel-2 reflectance, Sentinel-1 polarization, spectral indices and their corresponding textures were constructed. Thereafter, features with high permutation importance were retained. Three classification schemes with different training and test samples were, respectively, conducted. Finally, the VIIRS nighttime light data were utilized to refine the initial results. Manually collected samples and existing PV database were used to evaluate the accuracy of our method. The results revealed that the top three important features in detecting PV were the sum average texture of three bands (NDBI, VV, and VH). We found the classifier trained in highly urbanized coastal landscape with multiple PV types was more transferable (OA = 97.24%, kappa = 0.94), whereas the classifier trained in rural landscape with simple PV types was erroneous when applied vice versa (OA = 68.84%, kappa = 0.44). The highest accuracy was achieved when using training samples from both regions as expected (OA = 98.90%, kappa = 0.98). Our method recalled more than 94% PV in most existing databases. In particular, our method has a stronger detection ability of PV installed above water surface, which is often missing in existing PV databases. From this study, we found two main types of errors in mapping PV, including the bare rocks and mountain shadows in natural landscapes and the roofing polyethylene materials in urban settlements. In conclusion, the PV classifier trained in highly urbanized coastal landscapes with multiple PV types is more accurate than the classifier trained in rural landscapes. The VIIRS nighttime light data contribute greatly to remove PV detection errors caused by bare rocks and mountain shadows. The finding in our study can provide reference values for future large area PV monitoring.

**Keywords:** photovoltaic; land cover; image classification; transferability; Google Earth Engine

## 1. Introduction

Renewable energy has become the trend of modern power systems as it helps decrease greenhouse emissions and achieve carbon neutrality. Solar energy is very appealing to human society because it is abundant, inexhaustible and greenhouse gas free. Using solar panel arrays, photovoltaic (PV) technology can convert solar energy into electricity without any heat engine [1]. PV devices require a simple design, little maintenance [1] and declining costs in recent years. According to the International Energy Agency, the global solar PV generation increased 23% to reach 821 TWH in 2020 alone. China experienced a rapid expansion of PV installations, reaching one-third of global solar power by 2017 [2]. With the

carbon neutrality by 2050 or 2060 commitment from the European Union, the United States and China, PV installations are expected to grow in the near future.

There is increasing demand for the spatial distribution of PV installation. The distribution forms the basis to estimate PV power potentials and assess its ecological impact. For instance, the spatial distribution of PV modules is vital to estimate renewable potentials [3], as well as better operation and planning of power systems [4]. The evaluation of the impact of PV installations on climate [5], air temperature and energy balance [6] also relies on the spatial information of PV plants. Some researchers utilized PV information in landscape and urban planning as well [7].

Remote sensing can play an important role in detecting PV installation. Conventional methods, including household surveys and utility interconnection filings, are limited in their completeness and spatial resolution in collecting the distribution of PV plants [8]. On the contrary, remote sensing has the advantage in providing spatially explicit and time-series monitoring of land cover change. Some researchers proposed various methods to extract solar panels automatically from very high-resolution aerial images [9–11]. However, these data have limited spatial coverage and low temporal resolution. Medium resolution satellite images are appealing in this regard because they are freely accessible and make much more frequent observations of the Earth's surface. Recently, there are some studies utilizing such images to extract and monitor the development of PV. For instance, Kruitwagen et al. created the global inventory of PV from Sentinel-2 and SPOT 6/7 satellite images using deep learning models [12]. Unfortunately, the SPOT 6/7 images are not freely available for users from all countries. Some researchers mapped the distribution of PV plants using Landsat 8 OLI images in northwestern China, including Ningxia Province [13] and Golmud city, in China [14]. Furthermore, some researchers estimated the installation time of each PV unit and discovered the rapid expansion of PV in five northwestern provinces of China from 2007 to 2019 based on multi-temporal Sentinel-2 images [15]. Nevertheless, these studies conducted their analysis in remote areas characterized by low population density and landscape with little human intervention. Compared to rural areas, the PV deployment in urbanized areas are more likely to be distributed rather than centralized. Due to the limited land resources, there are more diverse PV types in coastal urban area such as cropland PV and water PV. Furthermore, the high spectral diversity of urban materials also adds to the difficulty of PV detection. The performance of PV detection using medium resolution satellite images in highly urbanized coastal areas remains to be explored.

Therefore, this study aims to detect PV installation in diverse landscapes using open multi-source remote sensing data. In particular, we aim to evaluate the transferability of PV detection between rural and urban coastal regions. We adopted the freely available Sentinel-1 SAR, Sentinel-2 multispectral and VIIRS nighttime light data, and performed our analysis in two study areas of China using the random forest classifier. Manually collected samples and existing PV database were used to assess the results. Results from this study can provide reference values for future PV mapping to a large spatial extent.

## 2. Materials and Methods

### 2.1. Study Area

To evaluate the impact of diverse land cover types on PV detection, this paper selects two study areas in China, as representative of rural and urban coastal areas, respectively. The first study area is the Gansu province (Figure 1), located in the northwest inland of China and far away from the ocean. Gansu has an area of 425,800 km$^2$, and a population of 25.02 million in 2021. It has various climates including subtropical humid monsoon, alpine cold and dry climate. Its average temperature is about 0~14 °C. The annual precipitation is about 40–750 mm mostly focused in the southeast. Gansu is one of the richest provinces in sunlight resources of China, with about 2600 h sunlight annually [16]. From Figure 1a, there are many natural landscape types in Gansu, including the Gobi, desert, plateau, rugged mountain and river valley. There are also some grassland and forests in the southeast part. Gansu is one of the bottom three Chinese provinces in terms of urbanization degree (~52%),

which can be observed from the VIIRS nighttime light imagery in Figure 1b. Cities are scattered on the land with many uninhabited areas.

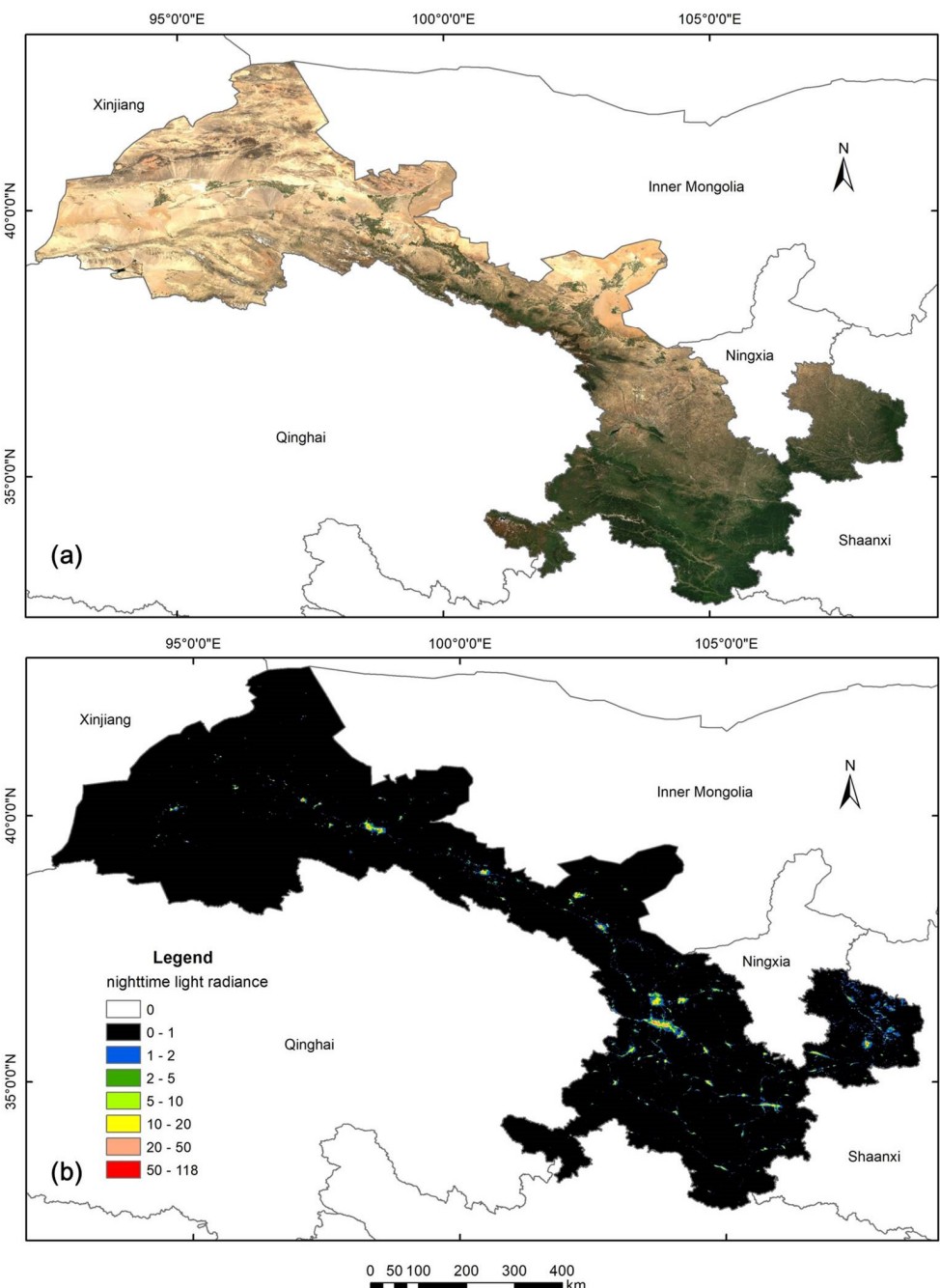

**Figure 1.** (**a**) The Sentinel-2 satellite true color composite image and (**b**) the VIIRS monthly average nighttime light radiance of Gansu Province.

In comparison, we choose the Zhejiang Province as the second study area. Zhejiang is located in the southeast coastal area of China (Figure 2). It has an area of 105,500 km², and a population of 64.57 million in 2021. As a result, the population density of Zhejiang is almost 10.4 times of that in Gansu at the province scale. Zhejiang has a subtropical monsoon climate with an annual average temperature of 15~18 °C. The annual average precipitation is 1200~2000 mm. The annual average sunshine hours is about 1170~1680 h [17]. Compared to Gansu, Zhejiang has a quite different landscape. Mountain forests and cropland covers about 61.17% and 12.23% of the land surface respectively in Zhejiang, which can be seen

from Figure 2a. On the remaining land, Zhejiang is highly urbanized with dense manmade buildings and impervious surface. The much higher urbanization degree and higher energy consumption of Zhejiang can also be observed from the much brighter nighttime light as shown in Figure 2b, especially in the northeast region. In addition, there are many rivers, lakes, ponds and reservoirs in Zhejiang (see Figure 2a, allowing for the construction of PV projects above water surface.

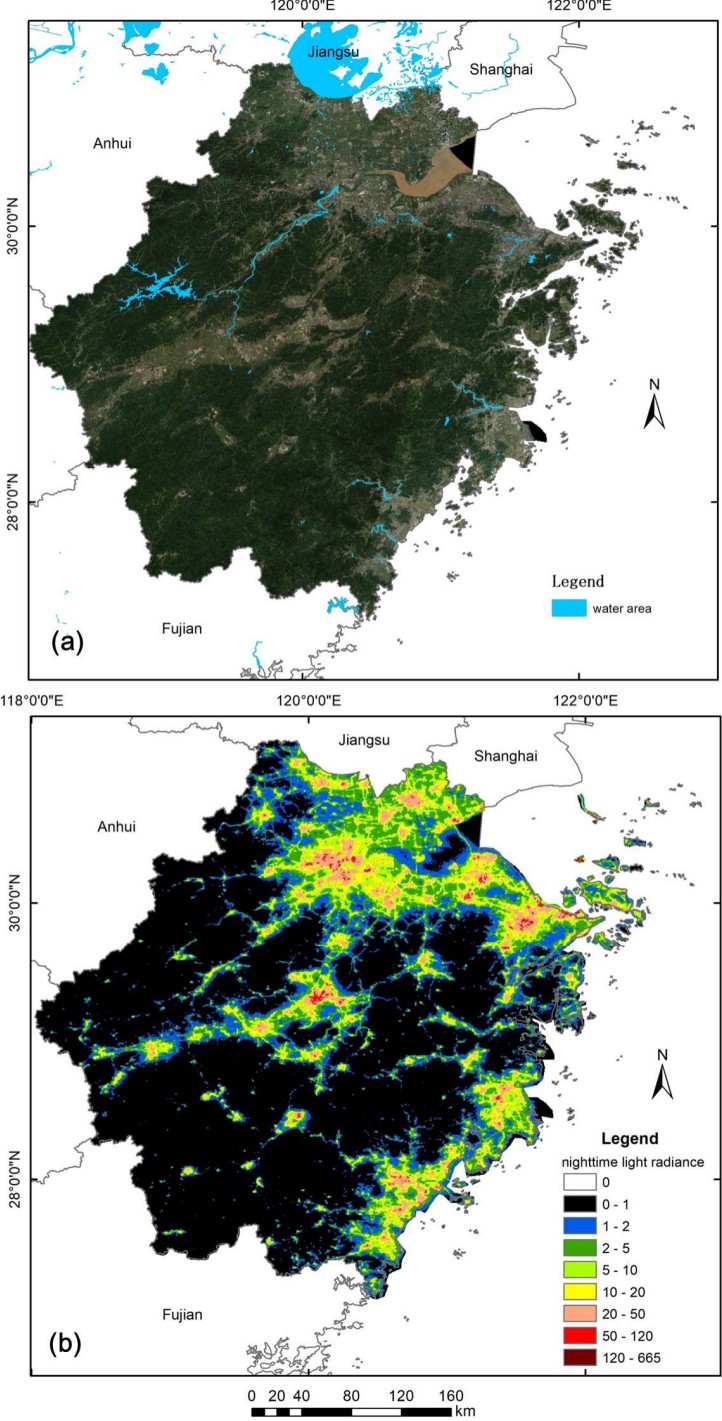

**Figure 2.** (**a**) The Sentinel-2 satellite true color composite image and (**b**) the VIIRS monthly average nighttime light radiance of Zhejiang Province.

## 2.2. Satellite Imagery

We employed multi-source satellite images including the Sentinel-2, Sentinel-1, and the VIIRS nighttime light data. In terms of satellites with multispectral sensors, there are Chinese satellites such as the Gaofen series. However, these data are not easily accessible over a large area and not published on the Google Earth Engine (GEE) platform, which means we cannot make use of the cloud computing power. Compared to Landsat, we chose Sentinel-2 due to its higher spatial resolution. The multi spectral instrument on Sentinel-2 has 13 spectral bands spanning from 0.44 to 2.2 μm. This covers the visible, near infrared (NIR) and short-wave infrared (SWIR) bands [18]. The spatial resolution varies from 10 m, 20 m to 60 m on different bands. We used the Sentinel-2 L2A surface reflectance product on the Google Earth Engine platform. We selected all images with a cloud percentage lower than 10% between April and October of 2021, including 2050 images in Gansu Province and 186 images in Zhejiang Province (The spatial distribution of the qualified observations count can be seen in the Supplementary Materials Figures S1 and S2). In terms of Sentinel-1, it provides all-weather data from a dual-polarization C-band Synthetic Aperture Radar (SAR) instrument [19]. We used the Sentinel-1 Ground Range Detected (GRD) product on GEE. This product has been calibrated and ortho-corrected, at a resolution of 10 m. Based on the suggestion from the previous study [20], we incorporated all the Sentinel-1 GRD products (VV/VH) between January and March (spring) of 2021, resulting in 543 images in Gansu and 88 images in Zhejiang. The third type of dataset we employed is the VIIRS nighttime light data. VIIRS acquires high quality nighttime images of the Earth [21,22]. It is usually used to map urban dynamics and serves as an indicator of socioeconomic activities [23–25]. We used the VIIRS monthly average cloud-free stray light corrected nighttime radiance product on GEE between April and October of 2021, resulting in 7 images in Gansu and Zhejiang. This dataset is used during the post-processing after initial PV detection. PV installations were more likely to locate proximate to human settlements rather than desolate and uninhabited area for reduction of transmission loss. Therefore, we utilized VIIRS nighttime light data to derive human settlements and remove PV detection errors in uninhabited area. More details can be seen in the Section 2.5.3.

## 2.3. Training and Test Samples for PV Detection

The study areas were classified into PV and non-PV (NPV) classes. We used stratified random sampling to collect training and test samples. First of all, 120 random points were generated in each study area to select non-PV classes. Because PV is a sparse class, none of the 120 random points were found to locate inside PV area in this experiment. Therefore for each NPV point, we manually drew a polygon around it with the help of visual inspection on the very high resolution satellite imagery on the Google Earth Pro platform, as demonstrated in Figures 3 and 4. Second, in terms of PV, we employed the existing PV database [12] as candidate areas and generated 120 random points inside these candidate polygons. Due to some misclassification errors in the dataset, there were two points outside the PV installation area. In such cases, we visually drew another polygon. Some examples of training and test sample polygons from Gansu and Zhejiang were presented in Figures 3 and 4. Afterwards, we split each group of 120 polygons randomly into 30 and 90, where 30 polygons were used as the test samples and 90 polygons were used as the training samples. As a result, the spatial distribution of the training and test samples for both PV and NPV objects in Gansu Province and Zhejiang Province were presented in Figures 3 and 4, respectively.

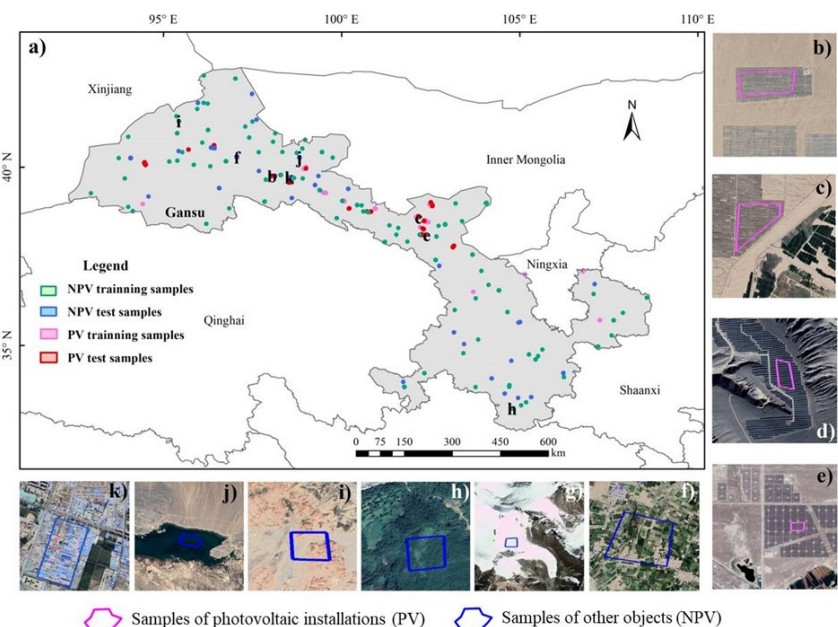

**Figure 3.** (**a**) The distribution of training and test samples for PV and NPV objects in Gansu Province, (**b**–**k**) are examples of the PV and NPV samples.

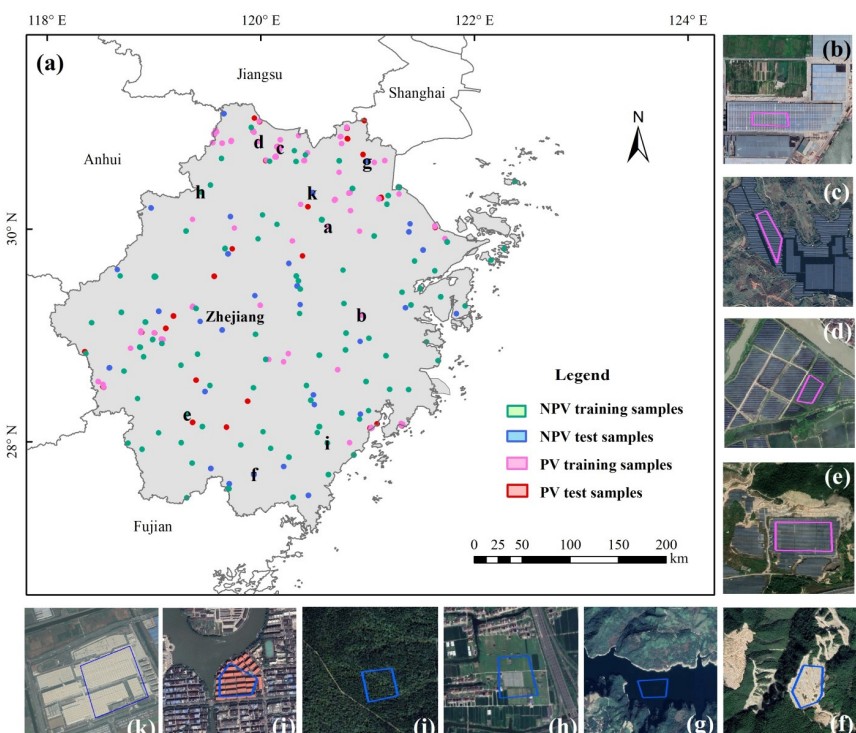

**Figure 4.** (**a**) The distribution of training and test samples for PV and NPV objects in Zhejiang Province, (**b**–**k**) are examples of the PV and NPV samples.

### 2.4. Existing PV Database for Consistency Evaluation

In addition to classification evaluation, we used three existing PV databases to evaluate the detection consistency of PV installations from this study with previous work. These data include (1) the Global Power Plant Database (GPPD) released by the World Resources Institute [26], (2) the published global PV database from the previous study [12], and (3) the published PV database in five northwestern provinces of China [15]. The number of existing PV installations in each study area is shown in Table 1 and details are as follows.

**Table 1.** Number of PV installations in existing published PV databases.

| Reference Database | Study Area | Number of PV Plants |
|---|---|---|
| GPPD | Gansu | 190 |
| Kruitwagen et al., 2021 [12] | Gansu | 253 |
| Xia et al., 2022 [15] | Gansu | 195 |
| GPPD | Zhejiang | 30 |
| Kruitwagen et al., 2021 [12] | Zhejiang | 173 |

First of all, the GPPD is an open-source power plant database. We used the GPPD v1.3.0 data (https://datasets.wri.org/dataset/globalpowerplantdatabase, last access 20 July 2022). In the dataset, each power plant is geolocated and there is an attribute describing fuel type [26]. All 220 PV data including 190 in Gansu Province and 30 in Zhejiang Province were extracted from GPPD. Second, we adopted the PV plants mapped in both Zhejiang Province and Gansu Province in the Kruitwagen's dataset [12]. It can be downloaded from https://zenodo.org/record/5005868#.YzJjZXZByUl (accessed on 20 July 2022). As introduced before, this dataset contains global PV mapped using Sentinel-2 and Spot 6/7 imagery till the end of 2018, each with a confidence attribute. We extracted only the polygons at the "A" confidence level so as to minimize errors within the database. In total, there are 253 PV polygons in Gansu and 173 PV polygons in Zhejiang. Lastly, we compared our results with the PV plants mapped in Gansu by the previous study [15]. This dataset is publicly available via https://code.earthengine.google.com/d6f17fa4fa44639db580d5f8b196fa5b (accessed on 20 July 2022). A total of 195 PV polygons were detected by Xia et al. using Landsat imagery in the year 2019.

*2.5. Method*

2.5.1. Preprocessing

Several preprocessing steps were conducted on the Sentinel-2 surface reflectance product. Firstly, the QA60 band was used in image cloud removal. Afterwards, these cloud-removed images were composited into one image using the 25 percentile of each pixel. We adopted the 25 percentile instead of neither median filtering nor minimum filtering so as to alleviate the effect of very bright pixels due to remaining cloud and very dark pixels caused by remaining cloud shadow. The resulted images in two study areas were presented in Figures 1 and 2, respectively. Since Sentinel-1 images were not influenced by clouds, we employed the median filtering to construct the composite image. Regarding the VIIRS nighttime light data, we also conducted median filtering to construct the annual composite in 2021.

2.5.2. Feature Construction

PV installations usually consist of many closely arranged monocrystalline or polycrystalline silicon panels. The strong absorption of crystalline silicon in visible bands lead to low PV reflectance in visible bands [27]. Moreover, there is a strong absorption around 2.2 μm of PV modules due to its hydrocarbon surface [27]. Therefore, we employed the reflectance of blue to SWIR bands including the B2, B3, B4, B5, B6, B7, B8, B8A, B11, and B12 to the feature space.

In addition to the raw reflectance, we adopted several spectral indices including the normalized difference built-up index (NDBI) [28], the normalized difference vegetation index (NDVI) [29], and the modified normalized difference water index (mNDWI) [30] to enhance the signature of these specific objects. We adopted the NDVI and mNDWI because we hypothesized the two indices might help detect PV installed above cropland/grassland or above water surface. These indices were also used in the previous study which used Landsat images to detect PV [31]. The equations for NDBI, NDVI, and mNDWI are as follows,

$$NDBI = (SWIR1 - NIR)/(SWIR1 + NIR) \tag{1}$$

$$\text{NDVI} = (\text{NIR} - \text{Red}) / (\text{NIR} + \text{Red}) \tag{2}$$

$$\text{mNDWI} = (\text{Green} - \text{SWIR1}) / (\text{Green} - \text{SWIR1}) \tag{3}$$

where SWIR1 and NIR are the short-wave infrared (1613.7 nm for S2A, and 1610.4 nm for S2B) and near-infrared band reflectance, while the Green and Red are the green and red band reflectance of Sentinel-2. As for polarization metrics, the previous study conducted in the Netherlands found that the VV and VH bands were among the top 10 important features for PV detection. Therefore we incorporated these bands as well [20].

Furthermore, because the layout of PV plants is generally highly directional to receive more sunlight, we take the image texture into account. Image texture inform us about the spatial arrangement of color or intensities in an image or selected region of an image [32]. Texture measures are usually derived from the gray level co-occurrence matrix (GLCM) [33, 34]. The previous study proved that adding texture features significantly improve the accuracy of PV extraction [13]. Thereby, for each feature we calculated a total of 17 textural metrics (see Table 2) on GEE [35,36]. As a result, there are a total of 270 features used. They are arranged in four groups listed in Table 3.

**Table 2.** The 17 types of textural features calculated in this study.

| Texture Symbol | Description |
| --- | --- |
| _asm | angular second moment |
| _contrast | contrast |
| _corr | correlation |
| _var | variance |
| _idm | inverse difference moment |
| _savg | sum average |
| _svar | sum variance |
| _sent | sum entropy |
| _ent | entropy |
| _dvar | difference variance |
| _dent | difference entropy |
| _imcorr1 | information measure of Corr. 1 |
| _imcorr2 | information measure of Corr. 2 |
| _diss | dissimilarity |
| _inertia | inertia |
| _shade | cluster shade |
| _prom | cluster prominence |

**Table 3.** All 270 features under assessment in this study.

| Feature Groups | Number | Specific Features |
| --- | --- | --- |
| S2 reflectance | 10 | B2, B3, B4, B5, B6, B7, B8, B8A, B11, B12 |
| Texture of S2 reflectance | 170 | Texture of B2, B3, B4, B5, B6, B7, B8, B8A, B11, B12 |
| Spectral indices & Texture | 54 | NDBI, NDVI, mNDWI Texture of NDBI, NDVI, mNDWI |
| S1 polarization & Texture | 36 | VV, VH Texture of VV and VH |

Note: S2 denotes Sentinel-2, while S1 denotes Sentinel-1.

With these numerous features, we conducted feature selection before classification to alleviate the effect of Hughes phenomenon that the accuracy of prediction is reduced due to the expansion of feature space over a finite sample size. In this study, we conducted feature selection within each feature group as listed in Table 3. More specifically, we exported the values of training samples from GEE to local hard drive. Afterwards, the permutation importance of each feature in each group was calculated using the python scikit-learn

library [37]. The permutation importance is defined to be the decrease in a model score when a single feature value is randomly shuffled [38]. Features with a high importance score were kept for the subsequent classification. Finally, the selected features from the four feature groups were combined into the final feature space.

### 2.5.3. Classification

After feature selection, we used the random forest (RF) classifier to classify the study areas into PV and non-PV classes. As a widely used machine learning algorithm, the RF classifier can successfully handle high data dimensionality and multicollinearity, being both fast and insensitive to overfitting [39]. The training and test samples for the RF classifier were as described in Section 2.3. Because we aim to evaluate the general applicability of PV detection in diverse landscapes, we investigated three classification schemes (see Table 4). In scheme 'GS_ZJ', we trained the classifier in Gansu and tested it in Zhejiang, while in scheme 'ZJ_GS', we trained the classifier in Zhejiang and tested it in Gansu. Lastly, in scheme 'GS+ZJ', we used all training samples in both provinces to train the classifier and used all test samples in both provinces to evaluate the accuracy. In theory, if the schemes "GS_ZJ" and "ZJ_GS" could achieve similar or comparable accuracy as "GS+ZJ", the detection of PV has high transferability. It is worth noting that, because different training samples were used in these classification schemes for feature selection as described in Section 2.5.2, different combinations of features may be selected.

**Table 4.** Three classification schemes under evaluation in this study.

| Scheme | Training Samples | Test Samples |
|---|---|---|
| GS_ZJ | Train_PV_GS + Train_NPV_GS | Test_PV_ZJ + Test_NPV_ZJ |
| ZJ_GS | Train_PV_ZJ + Train_NPV_ZJ | Test_PV_GS+ Test_NPV_GS |
| GS+ZJ | Train_PV_GS + Train_NPV_GS + Train_PV_ZJ + Train_NPV_ZJ | Test_PV_ZJ + Test_NPV_ZJ + Test_PV_GS + Test_NPV_GS |

Note: GS denotes Gansu, while ZJ denotes Zhejiang.

After the initial RF classification, we introduced a post-processing step by masking out suspicious PV beyond the area with human presence. In theory, PV will not be installed in the desolate and uninhabited area. In contrast, it is often built within or next to human settlements for reduction of transmission loss. Therefore, we refined the initial classification results by applying a mask indicating human activities. Satellite observed nighttime lights can provide a visual expression of human presence on the earth [40]. We extracted the urban extent by intensity thresholding (>3) on the VIIRS nighttime light radiance product and generated a buffer of 15 km around the extracted urban areas. The resulting mask was found to fully cover all existing PV polygons, as described in Section 2.4. Afterwards, only the identified PV located inside this mask were retained.

### 2.5.4. Accuracy and Consistency Evaluation

After classification, we first evaluated the accuracy of each classification scheme using the test samples. The commonly used overall accuracy (OA), user accuracy (UA), producer accuracy (PA), and kappa coefficient were calculated using,

$$OA = \frac{TP + TN}{TP + FP + FN + TN} \tag{4}$$

$$UA = \frac{TP}{TP + FP} \tag{5}$$

$$PA = \frac{TP}{TP + FN} \tag{6}$$

$$p_e = \frac{(TP + FN) * (TP + FP) + (FN + TN) * (TN + FP)}{(TP + FP + FN + TN)^2} \tag{7}$$

$$\text{Kappa} = \frac{\text{OA} - p_e}{1 - p_e} \tag{8}$$

where TP (true positive) is the number of PV pixels correctly predicted to be PV, and FP (False Positive) is the number of NPV pixels wrongly classified as PV. TN (true negative) is the number of NPV pixels correctly identified as NPV, while FN (False Negative) is the number of PV pixels wrongly classified as NPV. Out of the above-mentioned three classification schemes, we selected the most accurate one to map the spatial distribution of PV across two areas.

In addition, we compared our results with existing PV database. For each database listed in Section 2.4, respectively, we calculated the detection rate using,

$$p = \frac{\text{Num}_{predict}}{\text{Num}_{ref}} \times 100\% \tag{9}$$

where the $\text{Num}_{ref}$ is the number of PV polygons in existing PV database as presented in Table 1, and $\text{Num}_{predict}$ is the number of existing reference PV polygons which was also detected by this study. For each existing PV polygon, if this study classified more than 60% of its area into PV class, we will assign this polygon as detected.

## 3. Results

### 3.1. Feature Importance

The importance of the initial 270 features in four groups was sorted in descending order (see Figure 5). For visualization, those features beyond the top 15 important ones were not shown. In the Sentinel-2 reflectance group, B2 (Blue) and B11 (SWIR1) were consistently in the top four most important features in all three classification schemes, whereas three red edge bands including B5, B6 and B7 were always in the bottom four. Regarding the textures of Sentinel-2 reflectance, the "sum average" texture of B2 and B8A were more influential than other textures (see Figure 5b,f,j). In terms of spectral indices and the corresponding textures, the "sum average" texture of NDBI was the most crucial, far more important than mNDWI and NDVI (see Figure 5c,g,k). As for Sentinel-1 polarization features, the "sum average" texture of VH and VV were the top two features in all schemes. In general, the results reveal that textural features are very informative in detecting PV installation. As a result, we narrowed down the original 270 features to 13, 13 and 14 features, respectively, in three classification schemes (see Table 5).

**Table 5.** Selected features used in different classification schemes.

| Groups | ZJ_GS | GS_ZJ | GS+ZJ |
|---|---|---|---|
| S2 reflectance | B2, B3, B4, B8, B8A, B11, B12 | B2, B3, B4, B8, B8A, B11, B12 | B2, B3, B4, B8, B8A, B11, B12 |
| Texture of S2 reflectance | B2_savg, B11_savg | B8A_savg, B2_savg | B2_savg, B8A_savg |
| S1 polarization & Texture | VH_savg, VV_savg | VH_savg, VV_savg | VH_savg, VV_savg |
| Spectral indices & Texture | NDBI_savg, NDBI | NDBI_savg, mNDWI_savg | NDBI_savg, mNDWI_savg, NDBI_shade |
| Total Number | 13 | 13 | 14 |

Note: S2 denotes Sentinel-2, while S1 denotes Sentinel-1.

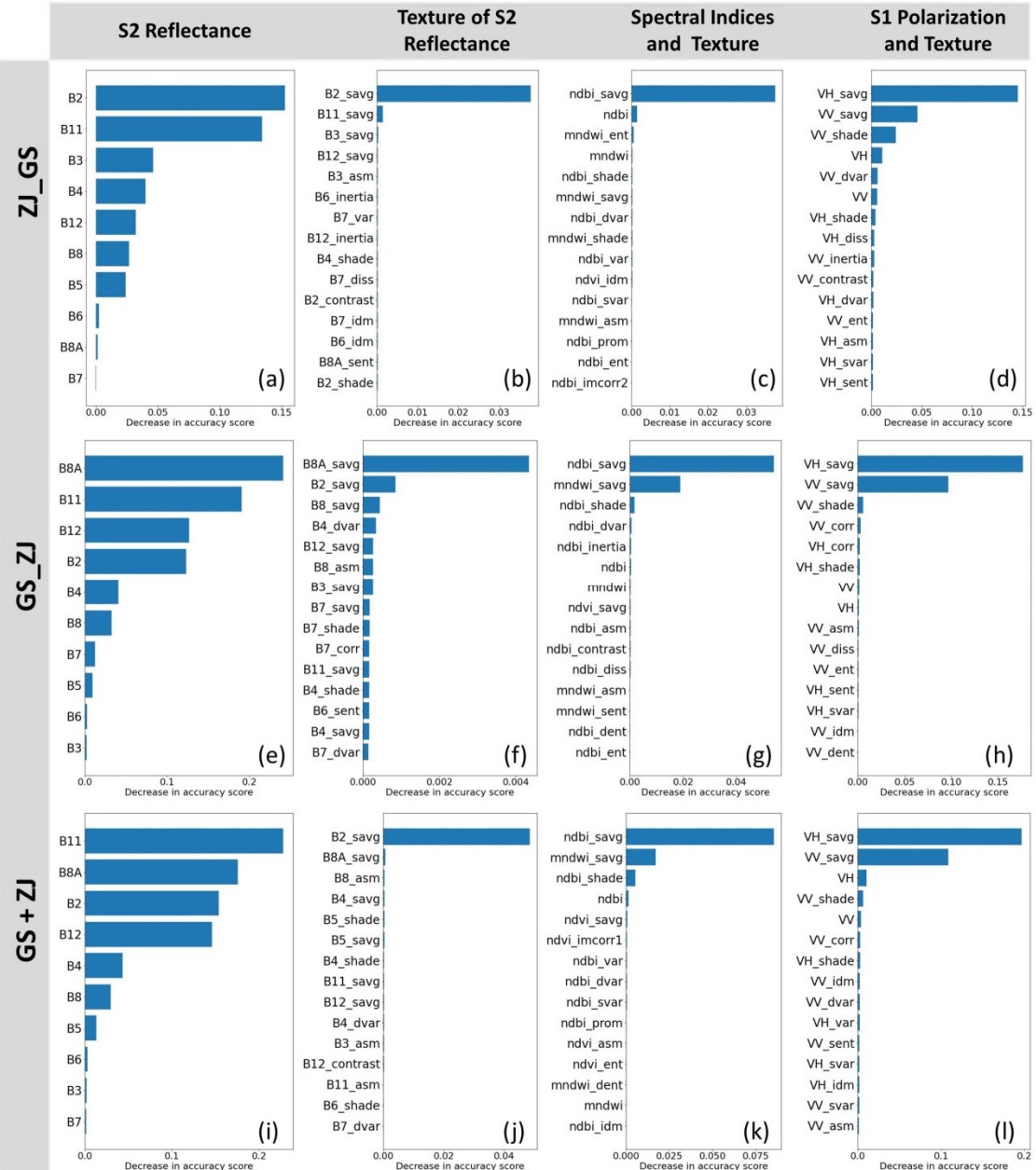

**Figure 5.** Permutation importance of different features in four feature groups in the three classification schemes. (**a–d**) are importance of features in classification scheme ZJ_GS; (**e–h**) are importance of features in classification scheme GS_ZJ; (**i–l**) are importance of features in classification scheme GS+ZJ.

### 3.2. Comparison among Classification Schemes

According to results in Table 6, the scheme "GS_ZJ" reached much worse results than the other two (OA = 68.84%, kappa = 0.44). The performance of the classifier trained in Gansu was far from satisfactory when tested in Zhejiang. In particular, the producer accuracy of PV (58.70%) and user accuracy of NPV (46.53%) were very low. Many NPV were wrongly detected as PV objects. On the contrary, the scheme "ZJ_GS" achieved much higher accuracy (OA = 97.24%, kappa = 0.94) using Zhejiang as the train area and Gansu as the test area. In the classification scheme "GS+ZJ", the highest accuracy was achieved as expected (OA = 98.90%, kappa = 0.98), when using training samples from both regions, which means more geographical heterogeneity was taken into account.

**Table 6.** Accuracy of different classification schemes.

| Scheme | OA | Kappa | UA_PV | PA_PV | UA_NPV | PA_NPV |
|--------|------|-------|--------|--------|--------|--------|
| GS_ZJ | 68.84% | 0.44 | 100% | 58.70% | 46.53% | 100% |
| ZJ_GS | 97.24% | 0.94 | 96.08% | 98.99% | 98.74% | 95.12% |
| GS+ZJ | 98.90% | 0.98 | 99.71% | 98.39% | 97.79% | 99.60% |

With the RF classifier on GEE, the resulting feature importance evaluation was shown in Figure 6. Out of the 14 selected features, the sum average textures of NDBI, VV and VH are the most important three features. This demonstrates the importance of textural and polarization information in addition to the Sentinel-2 reflectance in detecting PV.

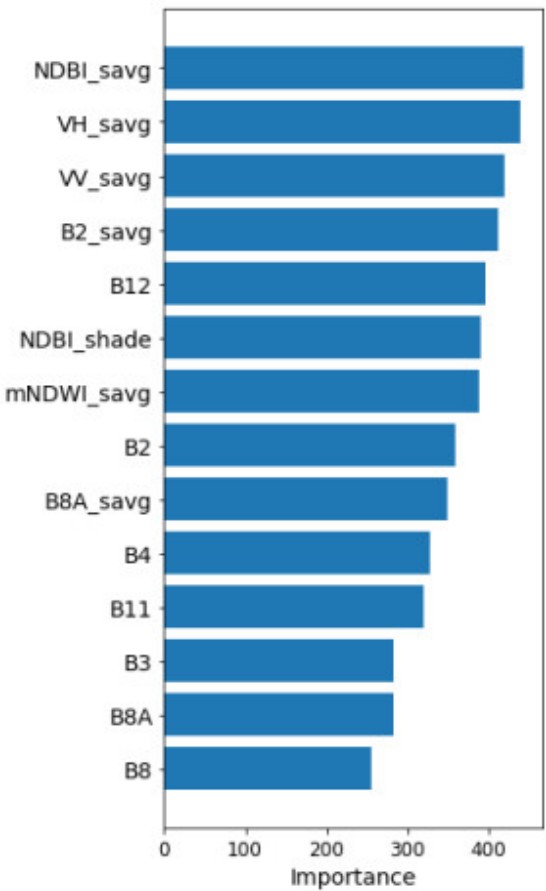

**Figure 6.** The importance of the 14 selected features in the RF classification.

### 3.3. Visual Inspection of Classification Results

According to Table 6, we selected the scheme "GS+ZJ" to detect PV in the two study areas. Example results of detected PV in Zhejiang Province are shown in Figure 7. For comparison, we outlined the PV detected by the previous study [12] in yellow. In general, this study detects most PV listed in Kruitwagen's database. Furthermore, this study has stronger detection ability of PV installed above water surface, which is often missing in Kruitwagen's database (see Figure 7c,f,i). We captured these fishery complementary PV projects which are rapidly developing in eastern China. However, a big shortcoming of this study is that there are some manmade buildings in urban areas such as the Jinhua Sports Center Stadium being incorrectly classified as PV (see Figure 7j–l).

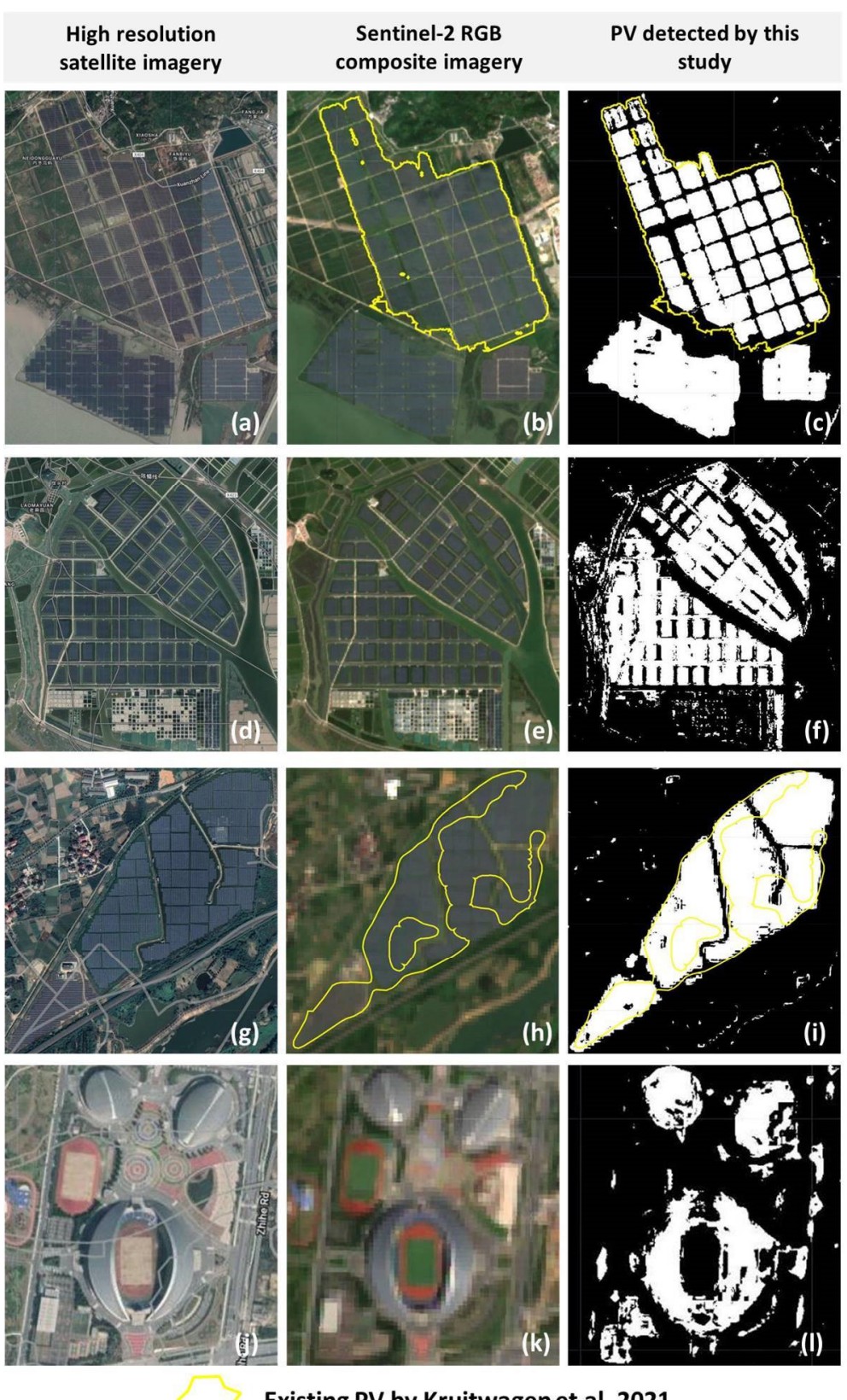

**Figure 7.** Classification results in Zhejiang Province (**a**–**c**) 121.32°E, 28.16°N; (**d**–**f**) 121.51°E, 29.16°N; (**g**–**i**) 118.71°E, 28.82°N; (**j**–**l**) 119.64°E, 29.05°N [12].

Some examples of detected PV in Gansu are shown in Figure 8. Unlike Zhejiang, which has a variety types of PV installed on different underlying surfaces, most PV detected in Gansu are constructed in the open Gobi desert. Generally they are well identified (see Figure 8). In some cases, hollow pixels occurred in the detected PV (see Figure 8f,i). In the satellite imagery, we can observe the higher brightness of some pixels in PV panels (see Figure 8d,e,g,h).

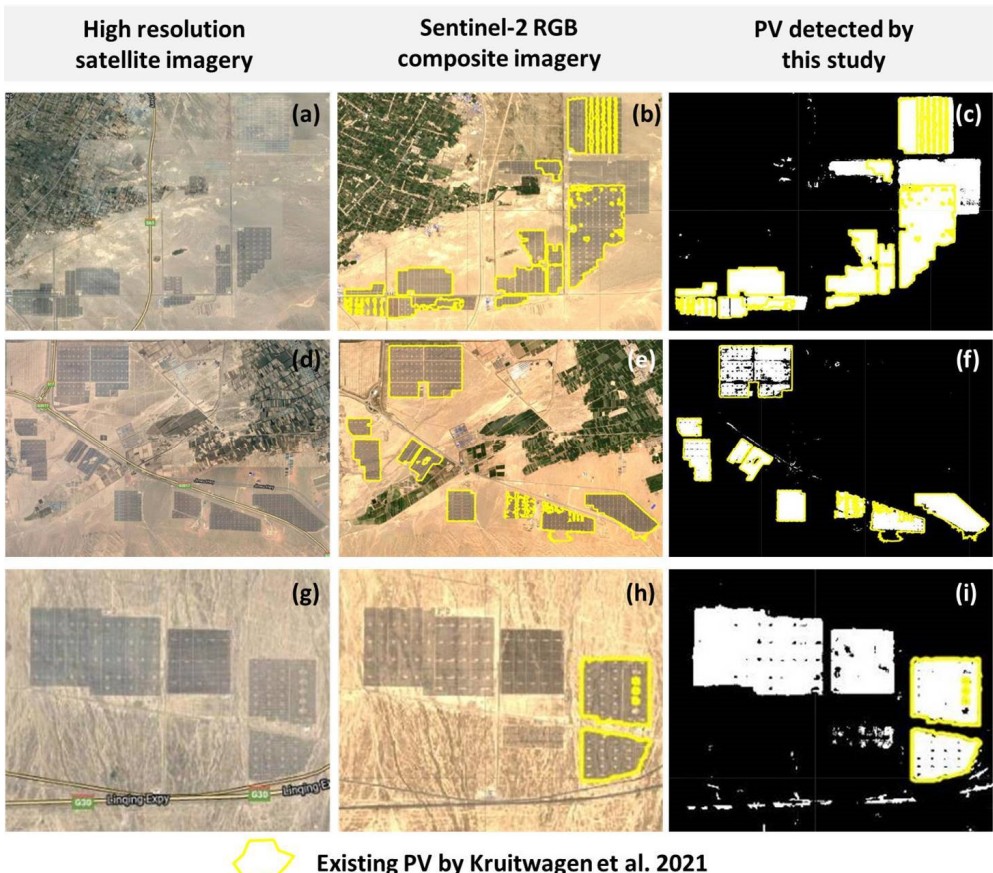

**Figure 8.** Classification results of PV stations in Gansu Province (**a–c**) 98.98°E, 39.97°N; (**d–f**) 102.37°E, 38.49°N; (**g–i**) 99.57°E, 39.29°N [12].

Moreover, there is a type of error which occurred in Gansu but is hardly found in Zhejiang, i.e., that some bare rocks or mountain shadows are misclassified as PV in the initial results (see Figure 9c). These errors often locate in natural landscape and far from human settlements. Using the post-processing step, many of these errors are removed (see Figure 9d). This proves that the nightlight time lights can provide beneficial values to refine the PV detection results.

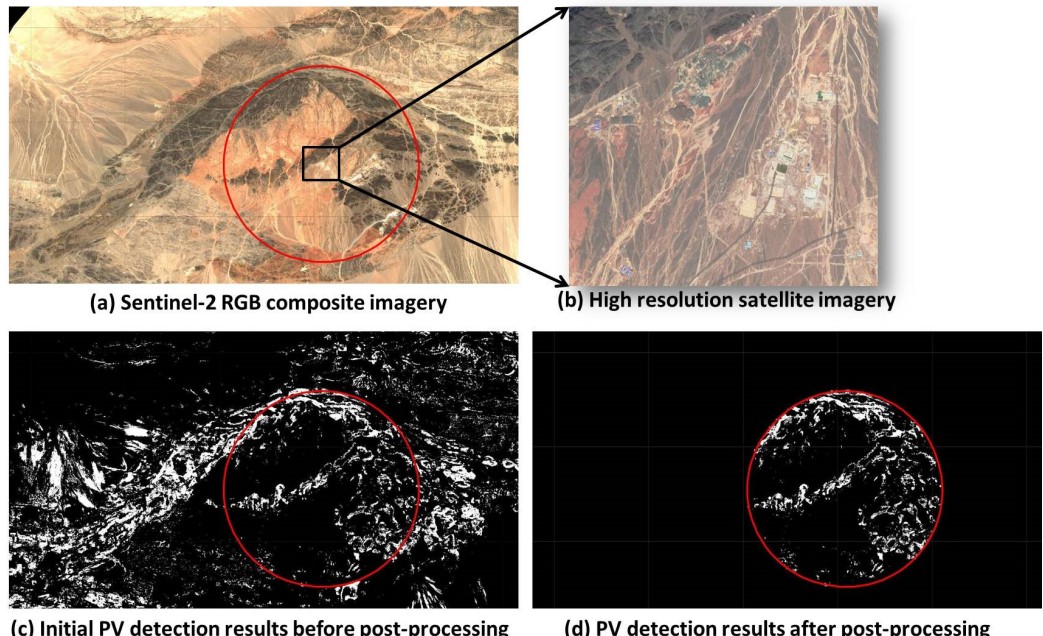

**Figure 9.** Bare rocks and mountain shadows misclassified as PV in Gansu Province (94.59°E, 40.90°N) (**a**) Sentinel-2 imagery, (**b**) manmade buildings in the center of the mask area, (**c**) the initial PV detection results before post-processing, (**d**) PV detected after post-processing.

### 3.4. Consistency Evaluation with Existing PV Databases

A quantitative comparison of PV detected by this study with existing PV databases listed in Section 2.4 is shown in Table 7. Overall, this study can produce comparable PV results. Aside from the GPPD database in Zhejiang, we recalled more than 94% PV whether compared to a remote sensing-derived database [12,15], or multisource-derived database [26].

**Table 7.** Consistency of our detection with existing PV database.

| Reference Database | Study Area | Number of PV | TP | Detection Rate |
|---|---|---|---|---|
| GPPD | Gansu | 190 | 183 | 96.31% |
| Kruitwagen et al., 2021 [12] | Gansu | 253 | 239 | 94.47% |
| Xia et al., 2022 [15] | Gansu | 195 | 190 | 97.44% |
| GPPD | Zhejiang | 30 | 26 | 86.67% |
| Kruitwagen et al., 2021 [12] | Zhejiang | 173 | 164 | 94.80% |

Upon further inspection, some undetected PV are shown in Figure 10. On the one hand, there are true omission errors [12,15], either due to the soiling issue (see Figure 10a–d) or due to unknown uncertainty (see Figure 10e–h). In some cases with cropland/grassland PV, the detection was only partial (see Figure 10i–l), perhaps due to spectral mixing caused by the dense vegetation under the PV during the growing season. On the other hand, there are some wrong omission errors caused by a change of land use. For instance, in Figure 10m–p, there is obvious land use change from February 2018 to January 2021. The previous PV detected by Kruitwagen et al. no longer existed in 2021. Furthermore, there are also some suspicious errors which require further fieldwork to verify. For example, we are not entirely certain to how interpret the undetected lower part of the yellow polygon in Figure 10q–t as PV, because they look very similar to plastic greenhouse cropland on the Google Earth Pro software. The azimuth direction of these objects is quite different from other south-facing solar panels. Further in-situ investigation may help us minimize such uncertainty.

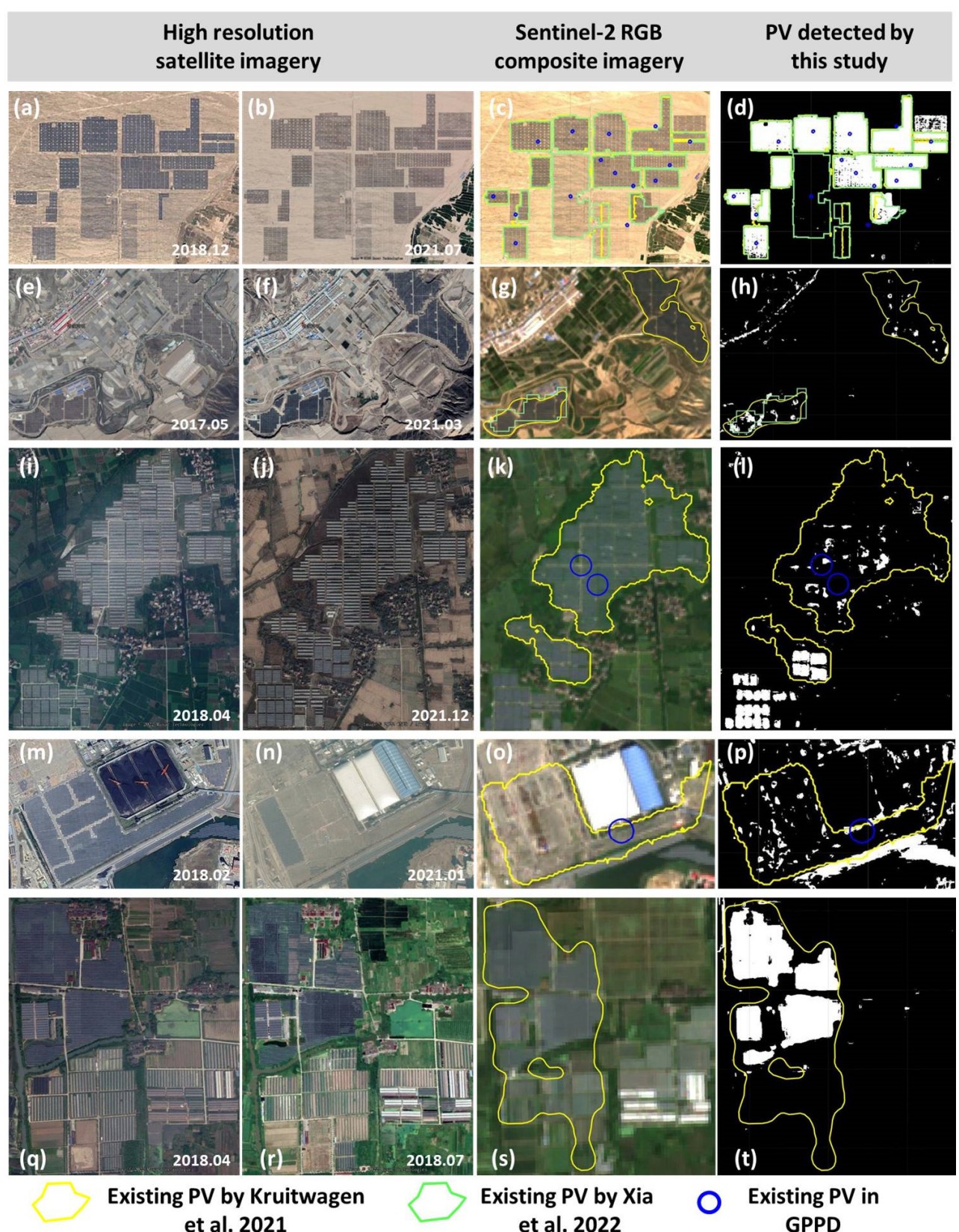

**Figure 10.** Undetected PV compared to existing PV database (**a**–**d**) 102.13°E, 38.60°N; (**e**–**h**) 104.58°E, 35.84°N; (**i**–**l**) 119.85°E, 30.95°N; (**m**–**p**) 121.09°E, 28.16°N; (**q**–**t**) 120.43°E, 30.71°N [12,15].

The final spatial distributions of PV extracted in Gansu Province and Zhejiang Province are shown in Figure 11. It is worth noting that there are still wrongly detected PV caused by urban roofing materials. We are conducting post-processing and visual confirmation before further release of our PV dataset.

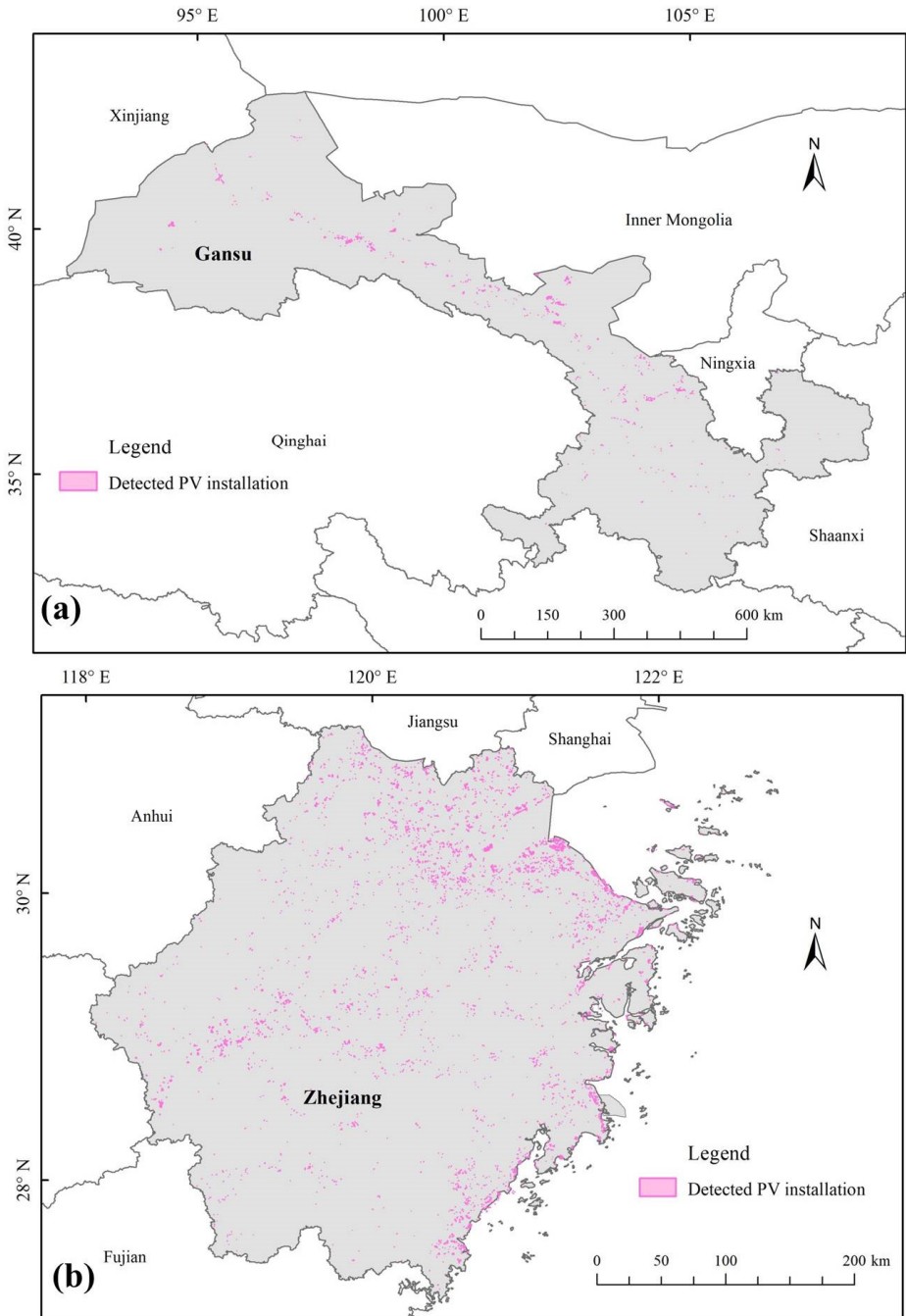

**Figure 11.** The final spatial distribution of PV extraction in (**a**) Gansu and (**b**) Zhejiang Province.

## 4. Discussion

### 4.1. Detection of PV Installations in Diverse Landscapes

From this study, we proved the feasibility of freely-available medium resolution satellite images for detecting PV in diverse landscapes. However, the detection accuracy was highly dependent on the training samples. In particular, the "ZJ_GS" classification scheme reached much better results than the "GS_ZJ" scheme, and was almost comparable to the "GS+ZJ" scheme (OA: 68.84% < 97.24% < 98.90%, Kappa: 0.44 < 0.94 < 0.98 in Table 6).

It is very likely because Zhejiang has more diversity of PV classes (such as cropland PV, bare land PV, grassland PV, reservoir PV etc.) so that the model has better generalization ability. This demonstrated that an RF model trained in urban coastal regions with multiple PV types such as Zhejiang is more transferable, whereas the classifier trained in rural landscape with simple PV types can be more erroneous. Similarly, Zhang et al. found that the rare PV samples collected from Dunnett's dataset [41] in eastern China severely limited the performance of the RF classifier to identify PV [31]. The importance of samples in the performance of the RF classifier was also strengthened in the previous review [39].

Compared to some previous work [13–15,42], we investigated detecting PV installation in various geographical regions instead of only remote areas such as northwestern China. Furthermore, in comparison with the study which mapped PV in the Netherlands using Sentinel imagery [20], there is much higher diversity in landforms as well as PV types in the Gansu and Zhejiang Provinces. Based on our results, we suggest training the classifier in diverse landscapes areas with multiple PV types for future work.

### 4.2. Importance of Multi-Source Remote Sensing

Our study demonstrates the importance of open multi-source remote sensing data to detect PV in diverse landscapes. The Sentinel-2 reflectance and corresponding spectral indices are important as expected because most solar panels are visually quite different from surrounding objects. However, due to the spectral intra-class variability and inter-class similarity, incorporating other features contributes to the detection. The importance of Sentinel-1 data is in compliance with the previous work [20]. Unlike studies which implemented object based classification without textural features [15,20], we employed pixel-based classification and proved the importance of textural features in detecting PV, in accordance with the previous work by Zhang et al. (2021). Unlike the previous study which only calculated textures on the NIR band [13], we calculated textures on the Sentinel-2 reflectance bands, Sentinel-1 polarization bands, as well as the spectral indices. As a result, we found that the "sum average" texture of NDBI, VV and VH all played great roles in detecting PV. As a matter of fact, with only the top three important features (NDBI_savg, VV_savg, VH_savg) from Figure 6, we were able to detect PV with a not-bad accuracy (PA_PV = 97.32%, UA_PV = 96.47%), though there were more misdetection of NPV as PV class. Another contribution of this study is that we make use of the VIIRS nighttime light data to quantify human activities. Their importance was obviously illustrated in the post-processing to refine the initial classification results as shown in Figure 9.

### 4.3. Accuracy Evaluation of PV Detection

Because PV detection is a highly label-imbalanced classification problem, there is a need to validate PV detection results from varying perspectives [4]. In accuracy evaluation, most studies rely on visual checking with the help of previous solar database to collect samples [13,14]. An alternative approach is to generate a preliminary classification result to limit candidate sample regions for visual check [11]. In some cases, researchers combined field survey with visual interpretation to ensure the sample quality [14]. Especially when we cannot decide whether the sample is PV or NPV, as shown in Figure 10q–t, a future in-situ survey can provide great value in reducing the uncertainty.

Consistency evaluation with existing PV database is also widely used. The existing dataset can be generated from statistical records, household surveys, volunteer annotation or satellite imagery. For instance, Xia et al. compared their results with the GPPD database [26]. They identified 90% of the PV listed in GPPD [15]. They also successfully identified an additional 235 PV power stations which were not in the database. Zhang et al. (2022b) successfully identified about 93.83% PV compared to Dunnet's database [41] and about 86.34% PV compared to Kruitwagen's database [12]. In this study, our results were verified by comparing with multiple existing PV database as well. It is worth noting that during our experiment, we found some NPV objects incorrectly labeled as PV in previous

databases. Such errors may lead to more errors in the subsequent analysis. Therefore, it is better to conduct a one-by-one inspection of each PV polygon before publishing it.

### 4.4. Limitation and Future Work

Currently, the main obstacles we find in mapping PV in this study are two types of errors, including the bare rocks and mountain shadows in natural landscape and the roofing polyethylene in urban settlements. These two errors caused over detection of PV thus requiring further post-processing and more research to be removed, so as to generate a high-quality database. Regarding the bare rocks and mountain shadows, similar commission errors were found in the previous work [15]. Assuming PV installations were proximate to human settlements, we removed many of these errors by a post-processing step with the nighttime light-derived area constraint as demonstrated in Figure 9. Similarly, in the previous study [12], the authors limited their search area of PV by dilating a global human population-density map with a 7 km buffer. The advantage of using the nighttime light data is that it can be more frequently updated than the global human population density map. Another type of post-processing involves filtering out PV pixels on locations with large or shady slopes [31]. For errors caused by urban roofing materials, previous studies pointed out materials such as roofing polyethylene [27], plastic-covered sheds [31] and solar thermal collectors [43] led to false detection of PV. The future release of hyperspectral imagery from the German EnMAP mission [44] may provide more spectral information to help us differentiate PV from these roofing materials. As the texture of VV and VH polarization bands were found as important features, further studies may also explore using other SAR satellite with higher spatial resolution such as Gaofen-3 [45,46] and Hisea-1 [47] for PV mapping. Furthermore, future studies may investigate the combination of satellite imagery with socioeconomic information such as the point-of-interest and social media check-in data to refine the initial classification results [48]. On the basis of the derived spatial PV distribution, it is also possible to use PV potential models combined with other input parameters such as solar radiation [49,50], cloudiness [51], and rooftop areas [52] to estimate energy output from PV modules and assess their contribution for carbon neutrality.

In addition, the open sharing of high-quality PV datasets across China is important. Such datasets can provide great potential in many applications to better support sustainable development goals. A recent study makes much progress in producing the latest map of PV distribution in China by the end of 2020, using Landsat imagery [31]. In addition, some researchers mapped the distribution of water PV (WPV) in China and found that WPV reached 165 km$^2$ in 2019. In the context of accelerating PV deployment in China, freely accessible and high quality PV database holds great value for energy policy formulation and environmental impact assessment [53].

## 5. Conclusions

Solar photovoltaics (PV) are rapidly expanding in China as a popular renewable energy technology. Medium resolution remote sensing (RS) plays an important role in monitoring the spatial distribution of PV. As China is a country with vast and diverse landscapes, the classifier trained in small areas may have poor performance, to a large extent. This study investigated detecting PV in diverse landscapes, taking the rural Gansu and the urban coastal Zhejiang Province in China as two representative areas. Using open multi-source RS including the Sentinel-2 multispectral, Sentinel-1 SAR, and VIIRS nighttime light data, we analyzed the transferability of the random forest classifier in detecting PV. Manually collected samples and existing PV database were used to evaluate the accuracy. The results proved the feasibility of medium resolution open RS data for detecting PV in various landscapes (OA = 98.90%, kappa = 0.98). However, the accuracy was highly dependent on the training samples. In particular, the RF classifier trained in highly urbanized coastal region with diverse PV types is more transferable (OA = 97.24%, kappa = 0.94), whereas the classifier trained in rural areas with simple PV types can be more erroneous

(OA = 68.84%, kappa = 0.44). Compared with the existing PV database, our method can recall more than 94% existing PV in most cases. Furthermore, our method has stronger detection ability of PV installed above the water's surface compared to existing PV databases. We found the top three important features for PV detection are the sum average texture of three bands (NDBI, VV, and VH). From this study, we found two main types of errors in mapping PV, including the bare rocks and mountain shadows in natural landscape and the roofing polyethylene materials in urban settlements. The VIIRS nighttime light data contributed greatly to remove errors caused by bare rocks and mountain shadows in the post-processing. Nevertheless, the diverse urban roofing materials are more complicated and warrant future research. The finding in our study can provide reference values for future large area PV monitoring.

**Supplementary Materials:** The following supporting information can be downloaded at: https://www.mdpi.com/article/10.3390/rs14246296/s1, Figure S1: Number of Sentinel-2 observations (cloud < 10%, April 2021–Oct 2021) in Gansu, Figure S2: Number of Sentinel-2 observations (cloud < 10%, April 2021–Oct 2021) in Zhejiang.

**Author Contributions:** Conceptualization, J.W., J.L. and L.L.; methodology, J.W. and J.L.; validation, J.W. and J.L.; formal analysis, J.W. and J.L. investigation, J.W. and J.L.; writing—original draft preparation, J.W.; writing—review and editing, J.L. and L.L.; visualization, J.W. and J.L.; supervision, L.L.; project administration, L.L.; funding acquisition, J.L. All authors have read and agreed to the published version of the manuscript.

**Funding:** This work was supported by the Jiangsu Natural Science Foundation under Grant [BK20200722]; the Natural Science Foundation of the Jiangsu Higher Education Institutions of China under Grant [20KJB420001]; and the Jiangsu Province Innovation and Entrepreneurship Doctor Program.

**Data Availability Statement:** Data and codes used in this study are available from the corresponding author on request.

**Acknowledgments:** We thank the editor and reviewers for improving the quality of our work.

**Conflicts of Interest:** The authors declare no conflict of interest.

## Abbreviations

| | |
|---|---|
| FN | false negative |
| FP | false positive |
| GEE | Google Earth Engine |
| GLCM | gray level co-occurrence matrix |
| GPPD | Global Power Plant Database |
| GRD | Ground Range Detected |
| mNDWI | modified normalized difference water index |
| NDBI | normalized difference built-up index |
| NDVI | normalized difference vegetation index |
| NIR | near infrared |
| NPV | non photovoltaic |
| OA | overall accuracy |
| PA | producer accuracy |
| PA_PV | producer accuracy of photovoltaic objects |
| PV | photovoltaic |
| RF | random forest |
| RS | remote sensing |
| SAR | Synthetic Aperture Radar |
| SWIR | short-wave infrared |
| TN | true negative |
| TP | true positive |
| UA | user accuracy |
| UA_PV | user accuracy of photovoltaic objects |
| VH | vertical transmit and horizontal receive |
| VIIRS | Visible Infrared Imaging Radiometer Suite |
| VV | vertical transmit and vertical receive |

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
