# Peer review of "Detecting Photovoltaic Installations in Diverse Landscapes Using Open Multi-Source Remote Sensing Data"

_remotesensing, doi:10.3390/rs14246296_

Round 1
Reviewer 1 Report
This paper is very interesting to use remote sensing technology to monitor solar photovoltaics (PV), but there are still many problems to be solved in the paper. To help authors revise the manuscript, I suggest to improve the paper on the following items:
1. Line 13-21 is too cumbersome, please reorganize this part.
2. There are so many existing Photovoltaic products (such as https://doi.org/10.5194/essd-14-3743-2022), please explain the innovation of this paper in detail.
3. Please add the available image spatial distribution for the number of satellite images, and please refer to Figure 2 in the paper "Using Landsat 8 imagery to map rice planting area in Northeast Asia, phenology-based algorithm and Google Earth Engine"
4. Figures 3 and 4 are suggested to be combined into one Figure, and the pictures corresponding to the sample points are marked. See “https://www.mdpi.com/2072-4292/14/21/5361”.
5. The results section only needs to express the study findings, no discussion or explanation is required, please revise.
6. Results please add the final spatial distribution of PV extraction in the Gansu and Zhejiang Provinces.
Reviewer 2 Report
The paper is focused on detection of PV stations in different landscapes (rural and urbanized coastal area) using freely accessible Sentinel-1, Sentinel-2 images and VIIRS nighttime light data product from Google Earth Engine.
The used RF classifier (incl. preprocessing steps) is more erroneous in rural areas compered to urbanized coastal region.
The proposed method has stronger detection ability of PV installed above water surface compared to existing PV databases.
Usage of VIIRS nighttime light data contributed greatly to remove errors caused by bare rocks and mountain shadows (rural landscape) in the post-processing.
The contribution of the paper is the finding that usage of VIIRS nighttime light data to contribute greatly to remove errors caused by bare rocks and mountain shadows (rural landscape) in the post-processing.
Comments:
Fig. 1 and 2: incorrect interval boundaries; 1 -2; 2-5 (1-1.9, 2-4,9); division of the coordinate grid; 0' 0'' is redundant
minor typos, marked in the text

Reviewer 3 Report
The manuscript entitled Detecting Photovoltaic Installations in Diverse Landscapes using Open Multisource Remote Sensing Data after Major Revision.
In my opinion, the subject of this work is relevant for the Journal Remote Sensing MDPI after Major revisions and some moderate corrections.
The topic of the paper is very interesting and important, especially in the content of green energy and Remote sensing, digital analysis in hope to find better locations for Photovoltaic Installations.
The Remote Sensing journal readers wants interesting and quality papers.
First, before all the paper has the next sections and sub-sections (i.e. Abstract, Materials and Methods, Study area, Satellite imagery, Training and test samples for PV detection, Existing PV database for consistency evaluation, Method, Preprocessing, Feature construction, Classification, Accuracy and consistency evaluation, Results, Feature importance, Comparison among classification schemes, Visual inspection of classification results, Consistency evaluation with existing PV database, Discussions, Detection of PV installations in diverse landscapes , Importance of multi-source remote sensing , Accuracy evaluation of PV detection , Limitation and future work , Conclusions, etc.).
First, I recommend that the authors add Section of Abbreviation before the section of Conclusion because of many short terms and many specific scientific words within text.
The section Abstract
The section of Abstract is overall good written but it necessary to extend with minimum one sentence which explain the main results of this research or how this research is can extend knowledge in this topic.
The section Introduction
I strongly recommend to the authors to add two valuable references which can support their findings. These references good explained calculations and various Remote Sensing techniques of analysis of satellite data.
- Valjarević, A., Popovici, C., Štilić, A. Radojković, M. Cloudiness and water from cloud seeding in connection with plants distribution in the Republic of Moldova. Applied Water Science, 12, 262 (2022). https://doi.org/10.1007/s13201-022-01784-3.
- Wu, J.; Fang, H.; Qin, W.; Wang, L.; Song, Y.; Su, X.; Zhang, Y. Constructing High-Resolution (10 km) Daily Diffuse Solar Radiation Dataset across China during 1982–2020 through Ensemble Model. Remote Sens. 2022, 14, 3695. https://doi.org/10.3390/rs14153695
Also, I recommend to the author to extend this section and add more about geographical position and Photovoltaic potential in common.
Section Materials and Method
Line 92 about 0~14 C°, please can the authors add precise data, it is very important for precise calculations. Also, this state can be cited.
In this section the author must explain why they used Sentinel-2 satellite data, there are other such Chinese satellite missions or Landsat for example?
Line 108 the data of insolation included values between t 1710~2100 hours. This state not cited? Also, the authors must explain meteorological data better (from which meteorological station, period of observation, errors of observation if any?
Figure 1.
The authors in this Figure stated that used true color composite image, can the author explain why did they not used false color composite it is may be better for visualization of potential photovoltaic fields?
Figure 2. The author explained very interesting value of night time radiance. The authors below of this map (Figure), can explain how is possible to use Google Earth Engine GEE method or any other to compare light pollution or (night radiance).
Line 129 the authors wrote about C-band Synthetic Aperture Radar (SAR). Can the authors add explanations of other kind of Synthetic Aperture Radar in C-band it will be good to compare?
Line 139 It is good to explain in this section because of readerships.
Section Training and test samples for PV detection
Line 143 the author stated the used 120 randomly points in this research, why this number of points?
Figure 3. Very valuable picture but I suggest to the authors to better sorted samples of polygons.
Samples of photovoltaic installations (PV) must be in one colon for example left, samples of other objects (NPV) in other colon right.
Figure 4. Interesting and valuable maps, but I strongly recommend to the authors to fix resolution of the maps. Also, all maps are good to have coordinates.
Line 165 The authors stated that used main data from this database e Global Power Plant Database (GPPD), but I think there is another big database, please explain?
Line 171 (This database cited in the text must be labeled in the references with precise date of downloading).
Section method
I suggest to the authors to change title on Methods and Materials or Materials and Methods.
Line 207 The authors must better explain why they are used NDVI (Normalized Vegetation Difference Index), because for this purpose is enough to analyzed NDBI (Normalized Difference Built-up Index) and Normal Difference Water Index (NDWI).
The section Feature construction
In this section the authors must add more sentences which described similar and before published research. The paper is very innovative, but must be covered by similar research.
I think that in this section authors can add more about Solar ecliptic and solar angles through the year. It is also important for photovoltaic potential.
The section Classification
This section presents one of the most important part of this research. This section is also very good written. But, I recommend to the authors to add more GIS methods. For example, is good to add any buffer analysis tools.
The section Discussion
In this section in Figure 8, the authors explained advance of High resolution imagery. Can the authors better explain comparison with Sentinel 2 RGB composite imagery and PV detection. I think that authors.
Did the authors found errors according the accuracy of methods or depend of errors produced by natural effects, please explain better?
The section Conclusion
The section must be extended with the more sentences
Why this research is important for renewable energy section in China?
Did the authors found the connection between Natural environment and Built up areas with photovoltaic potentials?
Please, add more main results in the section of Conclusion.
This paper has the strong potential to be published. The authors did a lot of things within this manuscript. The paper is very interesting and scientifically correct.
In the end, I recommend Major Revision.
Good luck to the authors
The Reviewer#2

Round 2
Reviewer 1 Report
The manuscript has been carefully revised in accordance with the comments and suggestions, and the presentation is more convincing in support of their findings.
Reviewer 3 Report
The manuscript entitled Detecting Photovoltaic Installations in Diverse Landscapes using Open Multi-Source Remote Sensing Data can be accepted. The authors answered all of my comments and corrected all of the mistakes within the text.
In my opinion, this manuscript can be accepted in its present form.
Sincerely,
The Reviewer #3
